# Promoting Fruit and Vegetable Consumption for Childhood Obesity Prevention

**DOI:** 10.3390/nu14010157

**Published:** 2021-12-29

**Authors:** Frans Folkvord, Brigitte Naderer, Anna Coates, Emma Boyland

**Affiliations:** 1Tilburg School of Humanities and Digital Sciences, Tilburg University, 5037 AB Tilburg, The Netherlands; 2Open Evidence Research, Open Evidence, 08005 Barcelona, Spain; 3Department of Media & Communication, LMU Munich, 80539 Munich, Germany; brigitte.naderer@ifkw.lmu.de; 4Department of Psychology, Institute of Population Health, University of Liverpool, Liverpool L69 3BX, UK; Anna.Coates@liverpool.ac.uk (A.C.); E.Boyland@liverpool.ac.uk (E.B.)

**Keywords:** children, food promotion, fruit and vegetables, marketing, health

## Abstract

Currently, food marketing for unhealthy foods is omnipresent. Foods high in fat, salt, and sugar (HFSS) are advertised intensively on several media platforms, including digital platforms that are increasingly used by children, such as social media, and can be bought almost everywhere. This could contribute to the obesity epidemic that we are facing. As the majority of children and adolescents do not eat the recommended amount of fruits and vegetables (F&V), which leads to chronic diseases, we need to change the obesogenic environment to a healthogenic environment. Reducing the marketing of energy-dense snacks to children and increasing the promotion of healthier foods, such as fruits and vegetables, may be an effective and necessary instrument to improve the dietary intake of children and reduce the risk of their experiencing some chronic diseases later in life. With this focused narrative review, we provide an overview of how children and adolescents react to food promotions and how food promotional efforts might be a useful tool to increase the attractiveness of fruit and vegetables. This review therefore contributes to the question of how changing the advertising and media environment of children and adolescents could help create a world where the healthy choice is the easier choice, which would reduce childhood obesity and improve children’s health, as well as to make the food system more sustainable.

## 1. Introduction

Children’s media environment is full of representations of food products, and several content analyses show that the majority of these foods lack nutritional value and that the food consumption patterns that are portrayed can have potentially harmful effects on children’s eating behavior [1,2]. For example, in children’s movies, nearly half of all food presentations showcase fast food, salty snacks, soft drinks or candy, often labelled as food and drink products that are high in (saturated) fat, salt, or sugar (HFSS), thereby contributing to the obesity epidemic as we currently know it [3,4]. In addition, mounting evidence suggests that when food is consumed and/or positively evaluated in these movies, then this product is significantly more likely to be a non-core, unhealthy food (i.e., HFSS) compared to a fruit or vegetable [3,4,5]. The YouTube videos of influencers popular with young people have been found to often feature unhealthy food cues, described positively with reference to social contexts [6]. Similarly, a content analysis of adolescents’ social media found that 67% of food presentations exclusively depict non-core foods, such as soft drinks, fast food, or sweets, and typically present oversized portions [2]. The same trend is observable for branded food presentation. Alruwaily et al. [1] found that more than 90% of child influencers’ YouTube videos presented placements for unhealthy branded items. In addition, food consumption on social media platforms, such as YouTube, is often portrayed in a rather extreme and distorted manner. For example, so called Mukbang videos (loosely based on the Korean translation for “eating broadcast”) typically showcase the overeating of large quantities of food [7].

There is ample evidence that the presentation of unhealthy foods is connected to children’s and adolescents’ unhealthy food choices (e.g., [8,9,10,11]). Adolescents might be particularly vulnerable to the long-term negative effects of unhealthy food promotion, as they are exposed to extensive marketing efforts on social media and are not typically covered by regulatory efforts [12]. Along these lines, a recent study showcases how adolescents who reported higher exposure to social media content which featured non-core foods (HFSS) were significantly more likely to report higher consumption of these foods. However, exposure to vegetable and fruit presentations does not seem to have the same positive effects [2]. HFSS foods have higher intrinsically rewarding properties that make them more “wanted” and “liked” than fruit and vegetables [3], thereby inducing unhealthy eating behavior among children and reducing their intake of healthier foods. Additionally, the promotion of HFSS foods is omnipresent and increases the rewarding value of these foods. Moreover, some studies showed that the promotion of fruit and vegetables affects the intake, although a recent systematic review showed that the evidence is inconclusive [12].

In order to improve children’s eating behavior, there is an urgent need to systematically test novel and effective methods of making fruit and vegetables more appealing to increase their intake among children [3,4,11,13,14,15]. Fruits and vegetables have been found to be good for health, where the strongest evidence for fruits was found for cardiovascular protection, with possible evidence for decreased risks of colon cancer, depression, and pancreatic diseases, while for vegetable intake, a reduction in the probability for colon and rectal cancer, hip fracture, stroke, depression, and pancreatic diseases was found [16]. The main aim of this focused narrative review is to give an overview of how children and adolescents react to food promotions and how food promotional efforts might be a useful tool to increase the attractiveness of fruit and vegetables in order to change our food environment from an obesogenic to a healthogenic environment. While there are many strategies involved in making fruit and vegetable desirable for children, such as health literacy programs in schools [17], or parents’ communication strategies and feeding practices [18], this review will focus on the role of food promotion. Thus, the review will contribute to the question of how changing the advertising and media environment of children and adolescents could help create a world where the healthy choice is the easier choice, which can reduce childhood obesity, improve children’s health, and make the food system more sustainable. In order to do so, we will first present the effects of marketing of unhealthy foods on children’s eating behavior, and subsequently show how the promotion of healthier foods could also effectively change behavior.

## 2. Child Development and Susceptibility to Food Promotion

### 2.1. Food Promotion and Children’s Susceptibility

Diets with high proportions of HFSS foods are marketed intensively and contribute to rising rates of childhood obesity [3,8]. Children with obesity are at increased risk of living with obesity as adults and developing a range of non-communicable diseases [19,20]. The World Health Organization asserts that the promotional strategies used in the pervasive and persuasive marketing of HFSS foods contribute significantly to the childhood obesity crisis [21]. Scientific evidence of children’s exposure to and the power of HFSS food marketing are herein discussed, followed by an examination of the evidence of its impact on children’s eating behavior and the theoretical underpinnings for interpreting this evidence.

In general, marketing to children affects food product sales in three ways [22]: (1) children have independent spending power (i.e., pocket money) which is often spent on snacks and confectionary, (2) children influence family spending (parents who go to the supermarket with their children are more likely to give in to their demands, which are predominately for branded HFSS products), and (3) children grow up to become adults, who are not only responsible for their own purchasing, but likely that of a family, making them a lifelong consumer. Thus, from a financial perspective, it is perhaps unsurprising that children are viewed as an important demographic target group by food marketers [23] and are preferentially targeted to a greater extent than any other group. Studies worldwide have monitored the types of products advertised in media popular with children and consistently find HFSS food marketing to be commonplace. For example, content analyses indicate that HFSS foods are more frequently marketed than healthy foods in the TV advertisements shown during children’s programming [24,25,26,27], in children’s programming and movies (i.e., product placements; [28,29]), and in digital media (including social media) where HFSS products account for 65–80% of all food marketing [1,30].

Digital media is now children’s preferred media [31] and various persuasive marketing techniques are employed by food brands to capture consumers’ attention in this space. For instance, many HFSS food brands (e.g., Coca-Cola, McDonalds) maintain social media accounts [32] and encourage users to interact with branded content through the use of competitions, coupons, monetary discounts, and interactive tools [31,32,33,34,35]. Brands also harness the connectedness of social media to further spread their promotional message and prompt users to “share” or “tag” branded social media posts with peers [36,37], and to incorporate branded products into their own social media posts. Such inducements to share are a powerful marketing tool, as these posts are perceived by others to be driven by consumers’ own experiences of a product (i.e., electronic word-of-mouth recommendations) [38] as opposed to a brands’ explicit and biased motive to promote a product [39,40,41].

Prompts from HFSS food brands to incorporate products into social media posts are successful [42]. For example, in Sweden, researchers analysed the content of adolescents’ (14-year-olds) Instagram posts for the frequency and nature of food cues displayed, and found that foods are frequently featured in this content (85% of images) and are mostly (68%) unhealthy. Furthermore, products are often presented as they would be in advertising, with branded foods clearly visible, positively described, and pictured in a fun and social setting [43]. Relatedly, “influencers” are individuals who have built audiences of followers through sharing user-generated content on social media [31], and brands will often pay or gift influencers with products to feature in their content. This marketing technique is known as “influencer marketing” [44]. Studies conducted in the UK, Europe, and Canada have indicated that the majority of foods promoted by influencers are classified as being unhealthy [2,3,43], and that this marketing significantly impacts young people’s food consumption [45,46].

In sum, unequivocal evidence demonstrates the negative impact HFSS food marketing has on children’s appetitive response. Multiple systematic reviews find empirically that exposure is associated with more positive attitudes towards, and preference for, HFSS foods, which leads to increased purchase requests (i.e., pester power), own purchasing (i.e., with pocket money), and actual consumption of such foods [3,8,10]. It is estimated that empirical exposure to HFSS food marketing is associated with children’s immediate increased intake of 30–50 kcals [3,8,10,47]. Furthermore, research finds that children (7–12 years) do not compensate for increased marketing-induced food consumption by reducing calorie intake at a later eating occasion [48]. Thus, over time, repeated food marketing exposure would likely lead to weight gain [3], especially because several studies have shown that children find it difficult to resist the influence of food marketing [3,49].

### 2.2. Resisting Food Marketing Influence

Psychological theories provide insights into the underlying mechanisms behind the promotional effects of food marketing on children’s eating behavior. More specifically, the reactivity to food cues in advertising model [3] asserts that visual food cues in advertising attract attention [50], trigger physiological responses (i.e., increased heart rate, salivation, skin conductance, and gastric activity) [51,52], and increase food cravings and intake [53]. The model also accounts for individual differences and message-level factors in reactivity to food marketing. For instance, children are more susceptible to the persuasive effects of advertising than adults [8], and endorsement effects (i.e., brand attitudes, purchase intention) are greatest when there is high congruence between an endorser and the product promoted [54,55].

In accordance with the persuasion knowledge model (PKM) [56], children under the age of 12 years are considered to be particularly vulnerable to the effects of marketing because they have not yet developed the level of cognitive ability required to understand the persuasive intent of advertising (i.e., that media content is produced for commercial gain and not a viewer’s entertainment) [22,23]. The ability to understand persuasive intent is argued to develop at around 12 years and should instigate scepticism and an increased resistance to persuasive attempts. However, research shows that even children as young as 8 years can display an understanding of persuasive intent and still respond positively to marketing, displaying a preference for the advertised brand compared to alternatives [57]. Consistent with these findings, empirical studies have demonstrated that advertising disclosures (visual disclaimers that notify the viewer of the persuasive intent of media content) have no protective effect on children’s appetitive response to HFSS food marketing, and can even increase the effect, with children consuming more of the advertised product when present [46,49].

Understanding how food marketing may affect individuals automatically, outside of conscious awareness, provides insights into the difficulties of resisting advertising effects, even for mature adults who have the cognitive ability to do so. Although according to the PKM, the recognition of persuasive intent is necessary to recognize and defend against advertising as it is conciously deemed appropriate [56], experience is also required to learn how to effectively defend against different types of advertising, which develops throughout life with exposure to different types of persuasive attempts. For adults, the best predictor of negative attitudes toward advertising occurs when individuals actively counterargue the messages in the ad [58]. Therefore, according to the PKM, because individuals are less aware of emotional effects and other automatic responses to advertising, these types of persuasive messages will be more effective than direct arguments about product benefits.

These automatic food marketing effects represent a form of “mental contamination” from external stimuli. According to Wilson and Brekke [59], several conditions are necessary to defend against these unwanted effects: the cognitive ability to resist; awareness of how one is influenced by the stimuli; and the motivation to resist. In addition, the food marketing defense model adapts this model to food marketing effects [60] and proposes four necessary conditions to defend against the influence of unhealthy food marketing: (1) the awareness of marketing stimuli, including conscious attention to a stimulus and an understanding of its persuasive intent; (2) the understanding of how one is affected and the outcome (e.g., liking the brand, consuming more food) and understanding how to effectively resist that influence; (3) the cognitive ability to resist, as well as available cognitive resources at the time of exposure; and (4) the motivation to resist. Relative to adults, children are likely less motivated to resist the influence of HFSS food marketing because they are driven more by taste and hedonic preference for these foods [61] than by long-term goals, such as health. In addition, they lack the executive control required to resist immediate gratification [62].

Considering the effects of marketing through this theoretical framework provides insights into how marketing influences children and adolescents, despite their understanding of persuasive intent and skepticism of marketing. This framework also explains how marketing can bypass conscious information processing and affect brand attitudes and behaviors automatically, and why effective emotional advertising is so powerful. Watching an ad that activates an emotional response will distract from considering the information (or lack of it) in the ad, thus deactivating a skeptical response. Therefore, ad liking is one of the strongest predictors of brand liking in children and adults [63]. This process can also explain how hidden forms of marketing, such as product placements embedded in movies, TV programming, games and music, and “influencer” marketing, can increase brand preferences, even when the viewer does not notice the brand message.

### 2.3. Development Vulnerabilities

Although advertising may be more efficient at creating brand schemas in younger children whose brains are rapidly developing, brain development occurs across one’s lifespan. In addition, adolescents’ still-developing cognitive abilities to delay gratification, increasing independence, use of media and brands to reflect burgeoning social identity, emotional sensitivity, and the means to purchase their own food, may make them uniquely vulnerable to the influence of unhealthy food marketing. For these reasons, public health experts have called on food companies to expand regulations. However, many of these developmental vulnerabilities continue through early adulthood. Life transitions, such as living independently for the first time, working full time, marrying, and having children, present opportunities, as well as risks, for adopting behaviors that promote health and prevent disease during young adulthood. Furthermore, extreme situations, such as the worldwide COVID-19 pandemic, might lead to significant shifts in eating behaviors (e.g., [64]). Health behavior patterns established during this early lifestage predict weight gain and lifetime chronic disease risk, including obesity, type 2 diabetes, hypertension, and cardiovascular disease [65]. The World Health Organization asserts that Government-led protection of all individuals should be made a priority [21] and that regulations should not only be designed to protect younger children. In addition, several member states within the European Union are advocating to stop the food marketing of unhealthy foods to children and promote healthier food instead.

## 3. Promoting F&V among Children

### 3.1. Eating Fruit and Vegetables (F&V)

Numerous studies have consistently shown that the dietary intake patterns of older children do not meet (inter)national standards (e.g., insufficient F&V and overconsumption of foods high in fat, sugar and salt [HFSS]), especially among children from low SES [66]. A European study showed that only 8.8% of children consume the recommended five servings of F&V a day [67], while a majority of the children consume too much HFSS foods. Unlike the consumption of F&V, HFSS foods lead to an automatic increase of activity in the brain’s reward system, thereby overruling homeostatic mechanisms, leading to an increased intake of these foods and, eventually, to overweight and obesity [68]. In contrast, eating a diet rich in F&V is essential for healthy growth and development [69], preventing unhealthy weight gain and obesity during crucial phases throughout people’s lifespan [70], protecting against many chronic diseases [71], and increasing mental well-being [72].

In addition to the intrinsic rewarding properties of HFSS foods, an effective mechanism to increase the consumption of a specific kind of food is through food promotion (a range of marketing techniques to add value to a food product and persuade people to consume this product, such as advertising on television or on social media [73]). Food companies frame their messages to prime children to focus on the hedonic and intrinsically rewarding aspects of HFSS foods [74]. Considering the effectiveness and success of promoting HFSS foods, the current article discusses whether the food promotion of healthier foods is also effective.

### 3.2. Scientific Evidence

Until now, a limited amount of research has suggested that it is challenging to increase the intake of F&V among children, especially over the long term [11,75,76,77,78,79]. However, there is mounting evidence for the effectiveness of the promotion of healthy foods over the short term, although this depends on multiple moderating factors (e.g., gender, BMI, ethnicity, and SES: for an overview, see [11]). Several limitations of exisiting studies have been established, such as the focus on the short term only, and limited funding, which meant that the studies were very generic, and they mostly included non-branded fruit and vegetables products [11]. F&V are less intrinsically rewarding than HFSS foods, considering the automatic physiological (e.g., saliva, increased activation in brain areas related to reward and food motivation systems) and psychological (e.g., craving, hunger, liking and wanting) reactivity that HFSS foods induce. F&V, on the other hand, miss these specific motivational capacities and incentives to automatically increase the willingness to consume these foods [11]. Additionally, the food industry uses incessant, sophisticated, and personalized advertising to effectively increase the hedonic and rewarding value of HFSS foods by modifying attitudes, emotions, intentions, and ultimately consumption behavior [11], and there is a lack of commercial imperative to promote F&V in the same way. Nonetheless, several studies have shown that exposure to F&V can be stimulated by simply exposing children at a very young age [80,81,82].

Most published studies focused on decreasing the reinforcing value of HFSS foods, while it is also important to investigate the potential of reinforcing the values of healthier foods, and assessing whether the reinforcing value also increases the long-term intake, which is the state of the art in this area of research. In addition, multiple studies have shown that lunch time at schools is a very promising and novel context to increase the intake of F&V because children increasingly spend their lunch time at school, and it is the most effective context to target children belonging to vulnerable groups [74]. Recently, an overarching theoretical model has been developed that explains and predicts how the food promotion of F&V works, using an eclectic synthesis of existing theoretical models from different disciplines and recent empirical evidence [4,11]. The four basic assumptions of the newly developed theoretical model, the promotion of F&V model, are that (1) by increasing the reinforcing value of F&V (e.g., liking and wanting) through effective food promotion techniques [80], (2) a reciprocal relation with eating behavior occurs, that, in time, (3) leads to a normalization of the intake of F&V (habitual formation). Furthermore, (4) individual and contextual factors (e.g., culture, BMI [83], SES [84], food fussiness [85], and parental feeding style [86]) determine individual susceptibility to food promotion and food acceptance. Testing the theoretical model will provide highly relevant insights into the effectiveness of the promotion of healthier foods, resulting in prevention strategies for overweight and obesity among children.

For example, a recent study investigated the effects of exposure to social media culinary videos on adolescents’ appetites [87] by exposing adolescents to either videos that featured the preparation of sweet snacks (HFSS) or snacks made from fruits or vegetables. Immediately following exposure, adolescents were measured for food choice behavior, liking of the foods, and intentions to eat and prepare the foods. The findings showed that those who watched the sweet snack preparation video had a reduced liking of fruits and vegetables and reduced odds of choosing a piece of fruit over a cookie. However, those who watched the fruits and vegetables video had a reduced liking of sweet snacks and higher intentions to prepare healthy snacks. Similar to these findings are those from a real example of healthy food marketing. In January 2019 and January 2020, a nationwide integrated marketing campaign “Eat Them to Defeat Them” was launched in the UK to inspire children to increase their consumption of vegetables [88]. In line with the PKM, overt attempts to influence children’s attitudes towards fruits and vegetables may result in a rejection of the message [56], and so the advert took a “fun” approach to consuming these foods rather than an “educational” approach about the benefits they can provide. This campaign harnessed a myriad of marketing techniques, including TV advertising, product placement in popular TV shows, digital media marketing, social media hashtags, celebrity endorsement, outdoor marketing, supermarket promotions, and a reward scheme in schools. An analysis found an average 2.3% weekly uplift of vegetable sales during the weeks that the campaign ran in 2019, equivalent to a retail value of GBP 16.2 million. Thus, when the scale and budget of a marketing campaign promoting healthy foods is comparable to that of HFSS foods, children can be encouraged to consume these foods. Considering F&V growers are often small businesses, diverse and fragmented across countries, it is extremely difficult for these producers to increase their budget on marketing campaigns. Collaboration between the different businesses is therefore essential to be able to collaboratively invest in marketing campaigns, such as the Dutch collaboration in the Fresh Produce Centre, where over 80% of total sales of F&V are associated. Another method is that the regional, national, or even supranational governments provide subsidies for these producers to invest in social marketing campaigns.

Although some studies have demonstrated positive outcomes of healthy food marketing, other research indicates that mere exposure to fruit or vegetables, for instance in product placement [89] or advergames [90], does not increase children’s and adolescents’ choice or intake of these foods. The influencer marketing of unhealthy foods increases children’s intake of these products [6], and so it could be hypothesized that exposure to fruits and vegetables that are presented by influencers in a positive way [2] could also encourage the consumption of these foods [11]. Indeed, children consider the opinions of the influencers they follow to be relatable and credible [91], and believe themselves to be influenced by their endorsements [3,4,82]. However, empirical findings indicate that short-term exposure to the influencer marketing of healthy foods does not encourage children to consume or choose these foods [15,92]. Similarly, some literature has demonstrated smaller-sized effects (or no effect) of celebrity endorsement of healthy foods, relative to unhealthy foods, on children’s appetitive response [93]. Thus, the promotion of fruits and vegetables is considered to not be as straightforward as the promotion of candy, fast food, or soft drinks, and that more persuasive tactics and repeated exposure may be needed [94]. Despite these findings, and due to the vast reach of influencers, influencer marketing is increasingly being used to promote health behaviors. For example, the WHO recently utilized influencer marketing to post accurate information about Coronavirus, and the UK government paid influencers to promote test and trace.

As demonstrated by studies that have employed eye-tracking technology, an explanation for the more powerful effects of HFSS food marketing compared to healthy food marketing could be because these foods more readily capture children’s automatic visual attention relative to fruit and vegetables [50,95]. Unhealthy food promotion on social media is more likely to be remembered and receive positive social responses, such as shares or comments, compared to the food promotion of healthy foods [96]. However, an examination of additional contributors, such as food literacy, suggest that social media can play an important role for adolescent eating, and thus there is an opportunity for health professionals to use social media in the promotion of fruits and vegetables to children and adolescents [2]. In addition to factors that relate to personal differences, contextual factors that relate to the food presentation itself might be important. Specifically, Binder et al., [94] suggest three pillars of persuasive strategies as part of their persuasive strategies for presenting healthy foods to children (PSPHF) typology: (1) composition-related characteristics, (2) source-related characteristics, and (3) information-related characteristics. For instance, composition effects studies suggest that more noticeable presentations, such as audio-visual presentations of fruit or vegetables, are more successful in eliciting healthy food choice effects in children compared to unimodal, audio presentations (8–11 years; [97]). Similarly, interaction presentations, where media characters actually handle food, arouse more visual attention in children (6–12 years; [50]) and thus when implemented in the food promotion of fruits and vegetables might lead to more healthy food choices.

Considering the source-related effects, studies suggest that majority cues [98] and what a media character represents are crucial. For instance, peer-led marketing, as opposed to influencer- or celebrity-led marketing, is suggested to be a stronger predictor of consumers’ eating behavior [99] and decisions to consume healthier foods [96]. In the context of influencer marketing, two studies suggest that the thematic background of the influencers—specifically, whether they relate to fitness and health—might be relevant. In this context, Folkvord and de Bruijne [15] showed that using a well-known fitness influencer led to higher healthy food brand attitudes and purchase intentions among adolescents (13–16 years) compared to a fictitious fitness influencer. De Jans et al. [100] found that using a social media influencer with a sedentary lifestyle, compared to an influencer with an athletic lifestyle, when promoting an unhealthy food leads to more healthy food choices in children (8–12 years old). This indicates that the negative consequences of a certain eating behavior seem to be effective in changing food preferences in children and adolescents. This suggests that information-related characteristics that relate to threat appeals are effective. However, employing threat narratives in the promotion of fruits and vegetables is connected to ethical concerns and might lead to unintended boomerang effects, such as reactance and forbidden fruit effects, and some studies also showed that health messages among children can backfire, resulting in more disliking of F&V [61].

Instead, promotional efforts might concentrate more on positive enforcement through the presentation of fruit and vegetables as rewards [94]. With regard to information-related characteristics, highlighting the positive aspects of fruits and vegetables that relate to emotional aspects [95] and highlighting the gains of eating healthy seems to be an effective avenue [94].

## 4. Discussion

### 4.1. Reversing the Obesogenic Environment: Creating a World Where the Healthy Choice Is the Easier Choice

The promotional strategies used in the pervasive and persuasive marketing of HFSS foods significantly contribute to “obesogenic food environments” [21]. These environments are characterized by the increased availability and affordability of unhealthy foods [101] and are considered to be a key driver of increasing childhood obesity globally. In the Western world, the unhealthy choice is often the easier choice. However, what if the healthy choice was the easier choice? The WHO contends that by tackling modifiable risk factors, such as limiting exposure to HFSS food marketing (e.g., regulation that restricts this marketing), providing necessary skills and information to encourage healthy food choices (e.g., via the promotion of fruit and vegetables, weight management services, and interventions offered by the NHS), and enhancing physical activity in schools and communities (e.g., initiatives such as the daily mile), childhood obesity rates and many non-communicable diseases could be delayed or even prevented [21].

Evidence of the detrimental effects of HFSS food marketing on children’s dietary health has led to calls from the WHO for countries to control, regulate, and restrict children’s exposure to this marketing (and its power to persuade) across *all* media (i.e., broadcast and non-broadcast) [19]. Earlier this year, a major report published by a joint commission between the WHO, the United Nations Children’s Fund (UNICEF), and the Lancet Commission [102] called for regulatory action to protect young people (up to the age of 18 years) from the marketing of unhealthy products, including tobacco, alcohol, and sugar-sweetened beverages. Few countries have responded by implementing government-led statutory legislation, while the majority have adopted self-regulatory codes of practice [103], codes developed and enforced by the food and advertising industries which pledge to only advertise healthy dietary choices to children [104]. However, in July 2020, the UK Government announced its intention to further restrict paid-for HFSS food advertising online, with new rules covering display and video advertising, and social media and influencer marketing, which are due to be implemented by the end of 2022 [105]. In addition, the European Commission has adopted the Digital Services Act in December 2020 in order to protect consumers and their fundamental rights online more effectively, establish a powerful transparency and accountability framework for online platforms, and to foster innovation, growth, and competitiveness within the single market [106]. However, it is not yet clear whether the regulations will be stringent enough to prevent marketers focussing their attention on less regulated areas. For example, the fact that brand marketing is not covered by the new rules may significantly limit their impact.

### 4.2. A Healthogenic Environment

A whole-systems approach is often cited as being fundamental to creating an environment whereby the healthy choice is the easy choice. While the personal environment, and thus family practices, are crucial to the socialization of a health-promoting environment [18], this is an area that is difficult to intervene in directly. Therefore, we need interventions from the public sphere to create a health-promoting environment that also affects the private space. For example, in addition to a ban on HFSS food marketing, radical interventions on price, product composition, labelling, and the availability of junk food are needed to see an impact. The Amsterdam healthy weight program [107] was launched in 2013 and is an example of such an approach. Program activities included a ban on marketing unhealthy foods at sports facilities, increased access to public drinking fountains and health ambassadors in different neighborhoods, “healthy playgrounds” (i.e., no access to junk foods), and the development of specific treatments for children living with obesity. Although there are no evaluations directly linking the program to changes to childhood obesity, the prevalence of children living with overweight and obesity in the city decreased from 21% in 2012 to 18.5% in 2015. Furthermore, in 2015, fewer children consumed sugar-sweetened beverages and exercised more than before the program was launched.

Recently, the COVID-19 pandemic has highlighted the importance of a robust digital infrastructure, and has led to calls for greater consideration of how digital processes and technologies may accelerate food system transformation for sustainable healthy diets. The current globalized food system cannot sustainably promote healthy people and a healthy planet [101]. However, based on the evidence presented in this article, the hope is that social media technologies can be used as part of a whole-systems approach to rethink the current status quo and use the tools at our disposal to promote healthy and sustainable choices for young people today and future generations [108].

## Data Availability

Not applicable.

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
