# Peer review of "Promoting Fruit and Vegetable Consumption for Childhood Obesity Prevention"

_nutrients, 2021, doi:10.3390/nu14010157_

Round 1

Reviewer 1 Report

The manuscript entitled ‘Promoting fruit and vegetable consumption for childhood obesity prevention’ presents interesting issue, however some corrections are needed

There are really important problems associated with the prepared manuscript, including:

  • The flaw of the presented manuscript is associated with the fact, that it presents a subjective review, not a systematic review. While the systematic review has a key role for broadening knowledge, the other reviews don’t have such role.
  • Taking into account, that the Materials and methods section is not presented (it should be added), without any specific information, it is hard to understand which studies were included into review and why. Authors did not present any key words, which were used during literature search, inclusion and exclusion criteria of references, information about the procedure of literature search conducted by them, number of chosen references, as well as information if some of them were excluded from the review and on the basis of which criteria. As a number of recent publications that are related to the issue were not included, it is a serious problem.
  • In this section Authors presented the information associated with influence of media environment on children. This section should be presented – what do we know and what is the background for this study. Some detailed information about other studies are necessary (international context – the situation in other countries should be presented). In the presented form – the title is not related to the introduction section. The good background should present the history of problem, the current knowledge and scientific "gap", and then authors should present how their study could fill this gap to justify the study.
  • The whole manuscript is associated with food ds high in fat, sugar and salt [HFSS] and fruit and vegetables (F&V), rather than promoting fruit and vegetable consumption (related to the dietary patterns)
  • Line 250 – there are some typos ‘especially in the long term[FF19]. ‘
  • The structure of this manuscript is unusual - 1. Introduction 2. Child development and susceptibility to food promotion 3. Promoting fruit and vegetables among children 4. Discussion. This structure must be improved.
  • Authors should emphasized the purpose of this review.

Author Response

Response Reviewer 1:

 C: The manuscript entitled ‘Promoting fruit and vegetable consumption for childhood obesity prevention’ presents interesting issue, however some corrections are needed

A: we thank the reviewer for the positive words. We will

C1: There are really important problems associated with the prepared manuscript, including:

The flaw of the presented manuscript is associated with the fact, that it presents a subjective review, not a systematic review. While the systematic review has a key role for broadening knowledge, the other reviews don’t have such role. Taking into account, that the Materials and methods section is not presented (it should be added), without any specific information, it is hard to understand which studies were included into review and why. Authors did not present any key words, which were used during literature search, inclusion and exclusion criteria of references, information about the procedure of literature search conducted by them, number of chosen references, as well as information if some of them were excluded from the review and on the basis of which criteria. As a number of recent publications that are related to the issue were not included, it is a serious problem.

A1: We agree that the manuscript would have serious flaws if we pretended it to be a systematic review, but in the abstract and introduction we already emphasize that we do not intended to conduct a systematic review but that we present a narrative review. A narrative review has a general approach and is to identify a few studies that describe a problem of interest, not necessarily all studies. Narrative reviews have no predetermined research question or specified search strategy, only a topic of interest, and do not have a systematic nature and follow no specified protocol.

C2: In this section Authors presented the information associated with influence of media environment on children. This section should be presented – what do we know and what is the background for this study. Some detailed information about other studies are necessary (international context – the situation in other countries should be presented). In the presented form – the title is not related to the introduction section. The good background should present the history of problem, the current knowledge and scientific "gap", and then authors should present how their study could fill this gap to justify the study.

A2: We agree with the reviewer and have revised the introduction accordingly.

C3: The whole manuscript is associated with food ds high in fat, sugar and salt [HFSS] and fruit and vegetables (F&V), rather than promoting fruit and vegetable consumption (related to the dietary patterns)

A3: According to the authors the bulk of the evidence lies in how effective unhealthy food promotion is, which we also reflected in the manuscript, seeking at the same time to emphasise emerging evidence that health promotion can also be effective

C5: Line 250 – there are some typos ‘especially in the long term[FF19]. ‘

A5: We apologize for this mistake and have revised it.

C6: The structure of this manuscript is unusual - 1. Introduction 2. Child development and susceptibility to food promotion 3. Promoting fruit and vegetables among children 4. Discussion. This structure must be improved.

A6: We agree that the structure is unusual, that is also why we choose to write a review article, instead of a research article. This is typical for a narrative review.

C7: Authors should emphasized the purpose of this review.

A7: We have emphasized this now better in the introduction.

Reviewer 2 Report

Folkvord et al Promoting F&V

This is a reasonably well-written narrative review that has novel value and is likely to be of interest to readers of Nutrients. It is better at analysing the problem than suggesting a practical solution. 

I am not sure that the numerical referencing is always correct.

Abstract

Replace “consequently” with “this could contribute to….”  i.e. use cautious language that does not imply causation (unless there is evidence).

The authors, correctly, attribute F&V consumption to reduced risk of some chronic diseases (in later life).  This should be included in the abstract.

Introduction

L31 delete “their” and replace with “children’s”

L%% delete ‘straightforward”.  Do the author’s mean “positive” effects? 

L61 I think Ref 13 should be Ref 12?

L131 rephrase as “…..empirically, that exposure….

L140 “specifically”

L150 delete “asserted” and replace with “considered”

L182 Does this model consider sensory preference?  There is evidence (as alluded to elsewhere) that this is key to children’s food choices.

L187 Ref 59 doesn’t seem to be the correct reference.  See Cox, D.N., et al., Sensory characteristics of vegetables consumed by Australian children. Public Health Nutrition, 2021: p. 1-12.

L189-190 But how does the model apply to children, the focus of the current review?

L211 typo

L213 Do the authors mean “early” life-stage?

Section 3.1 Seems to be a repetition of what has been said before without much being said about eating F&V.  Delete and include anything not previously said in earlier sections.

L250 what is ref FF19?  Reviews of promotion of F&V to children are needed here e.g.

Evans, C.E., et al., Systematic review and meta-analysis of school-based interventions to improve daily fruit and vegetable intake in children aged 5 to 12 y. The American journal of clinical nutrition, 2012. 96(4): p. 889-901.

Hendrie, G.A., et al., Strategies to increase children's vegetable intake in home and community settings: a systematic review of literature. Maternal & child nutrition, 2017. 13(1): p. e12276.

Also reference to previous social marketing F&V campaigns and reviews of their effectiveness and limitations (e.g short-term; funding; generic; non-brand etc. )

L275 There is evidence that exposure to the food stimuli itself is key (not just “promotion” e.g.

Bell, L.K., et al., Supporting strategies for enhancing vegetable liking in the early years of life: an umbrella review of systematic reviews. The American journal of clinical nutrition, 2021. 113(5): p. 1282-1300.

Bell, L.K., et al., Identifying opportunities for strengthening advice to enhance vegetable liking in the early years of life: qualitative consensus and triangulation methods. Public Health Nutrition, 2021: p. 1-39

Whilst I appreciate the focus of the current review is ‘advertising”, the authors need to locate it among other strategies, given that there is evidence that many strategies are needed simultaneously.

L275 Reference to mere exposure is needed here (Zajonc)

L277-279 these factors drive food acceptance

L303 Who pays for such promotion given that F&V growers are often small businesses, diverse and fragmented?  Whose role or responsibility is it?  In Australia, whilst there is an R&D levy of vegetable growers there is no marketing levy (see  https://www.horticulture.com.au/ hence there is limited funding for promotion.  How may this be overcome?  What happens in other jurisdictions? How has generic promotion worked e.g. Produce for Better Health in the USA?

L327-328 What is “remembered”? - could the same attributes/ stimuli remembered pertain to F and V?

L349 What kind of ‘experts”?

L358-363 Health messages among children can be counter-productive, leading to dislike (see Cox et al, 2021)

Section 4.1 There is no explanation of how multi-national social media companies can be regulated.  There is currently a strong narrative on their power and negative effects (misinformation, harm. hatred etc. e.g. revelations from former Facebook employees) but national governments have limited power (and/or willingness) to regulate or interfere in markets (depending upon their political ideology)

Author Response

Response Reviewer 2:

C: This is a reasonably well-written narrative review that has novel value and is likely to be of interest to readers of Nutrients. It is better at analysing the problem than suggesting a practical solution. 

A: We thank the reviewer for the positive words.

C1: I am not sure that the numerical referencing is always correct.

A1: We will check the references again to make sure they are correct.

C2: Abstract.

  • Replace “consequently” with “this could contribute to….”  i.e. use cautious language that does not imply causation (unless there is evidence).
  • The authors, correctly, attribute F&V consumption to reduced risk of some chronic diseases (in later life).  This should be included in the abstract.

A2: Revised accordingly.

C3: Introduction

  • L31 delete “their” and replace with “children’s”
  • L%% delete ‘straightforward”.  Do the author’s mean “positive” effects? 
  • L61 I think Ref 13 should be Ref 12?
  • L131 rephrase as “…..empirically, that exposure….
  • L140 “specifically”
  • L150 delete “asserted” and replace with “considered”

A3: Revised accordingly.

C4: L182 Does this model consider sensory preference?  There is evidence (as alluded to elsewhere) that this is key to children’s food choices.

A4: No the model does not include the sensory preference of children. In the rest of the manuscript we emphasized that the predetermined sensory preference of children is a main reason why food marketing for unhealthy foods is so effective.

C5: L187 Ref 59 doesn’t seem to be the correct reference.  See Cox, D.N., et al., Sensory characteristics of vegetables consumed by Australian children. Public Health Nutrition, 2021: p. 1-12.

A5: We have checked this reference and we believe it is the correct reference. Nonetheless, we will include the reference suggested by the reviewer as well.

C6: L189-190 But how does the model apply to children, the focus of the current review?

A6: We apologize for this mistake, it should have been children and adolescents.

C7: L211 typo

A7: Revised.

C8: L213 Do the authors mean “early” life-stage?

A8: Correct, revised accordingly.

C9: Section 3.1 Seems to be a repetition of what has been said before without much being said about eating F&V.  Delete and include anything not previously said in earlier sections.

A9: We have revised this section accordingly.

C10: L250 what is ref FF19?  Reviews of promotion of F&V to children are needed here e.g.

A10: Revised accordingly, we apologize for the mistake. We have included the references that were suggested.

C11: Also reference to previous social marketing F&V campaigns and reviews of their effectiveness and limitations (e.g short-term; funding; generic; non-brand etc. )

A11: We thank the reviewer for these suggestions and have included the references in the manuscript. In addition, we have listed some of the limitations of the studies in the manuscript.

C12: L275 There is evidence that exposure to the food stimuli itself is key (not just “promotion” e.g.

Bell, L.K., et al., Supporting strategies for enhancing vegetable liking in the early years of life: an umbrella review of systematic reviews. The American journal of clinical nutrition, 2021. 113(5): p. 1282-1300.

Bell, L.K., et al., Identifying opportunities for strengthening advice to enhance vegetable liking in the early years of life: qualitative consensus and triangulation methods. Public Health Nutrition, 2021: p. 1-39

A12: We thank the author for these suggested papers. We have now included the references in the manuscript and discussed the direct link to exposure and the intake.

C13: Whilst I appreciate the focus of the current review is ‘advertising”, the authors need to locate it among other strategies, given that there is evidence that many strategies are needed simultaneously.

A13: We completely agree with this comment, as advertising is just one out of many strategies that might make fruit and vegetable more desirable to children. We now added explicit mentions to school literacy programs and parental communication and feeding practices in the introduction. We circle back to both of these strategies in our discussion section.

C14: L275 Reference to mere exposure is needed here (Zajonc)

A14: Included.

C15: L277-279 these factors drive food acceptance

A15: We have revised accordingly.

C16: L303 Who pays for such promotion given that F&V growers are often small businesses, diverse and fragmented?  Whose role or responsibility is it?  In Australia, whilst there is an R&D levy of vegetable growers there is no marketing levy (see  https://www.horticulture.com.au/ hence there is limited funding for promotion.  How may this be overcome?  What happens in other jurisdictions? How has generic promotion worked e.g. Produce for Better Health in the USA?

A16: We agree with the reviewer this is an immense challenge, we have addressed this in our manuscript and included the following:

‘Considering F&V growers are often small businesses, diverse and fragmented across countries, it is extremely difficult for these producers to increase their budget on marketing campaigns. Collaboration between the different business is therefore essential, to be able to collaboratively invest in marketing campaigns, such as the Dutch collaboration in the Fresh Produce Centre, where over 80% of total sales of F&V are associated. Another method is that the regional, national, or even supranational governments provide subsidies for these producers to invest in social marketing campaigns.’

C17: L327-328 What is “remembered”? - could the same attributes/ stimuli remembered pertain to F and V?

A17: We do not think so, as is also shown in the study of Murphy, G., Corcoran, C., Tatlow-Golden, M., Boyland, E., & Rooney, B. See, like, share, remember: Adolescents’ responses 608 to unhealthy-, healthy-and non-food advertising in social media. International journal of environmental research and public health 609 (2020), 17(7), 2181.

C18: L349 What kind of ‘experts”?

A18: We have removed this sentence.

C19: L358-363 Health messages among children can be counter-productive, leading to dislike (see Cox et al, 2021)

A19: We agree and have included this in the manuscript.

C20: Section 4.1 There is no explanation of how multi-national social media companies can be regulated.  There is currently a strong narrative on their power and negative effects (misinformation, harm. hatred etc. e.g. revelations from former Facebook employees) but national governments have limited power (and/or willingness) to regulate or interfere in markets (depending upon their political ideology)

A20: We agree that it is extremely challenging to regulate social media companies. At the same time, the European Commission is trying to find ways to do so. We have now included some explanation for this:

In addition, the European Commission has adopted the Digital Services Act in December 2020, in order to protect consumers and their fundamental rights online more effectively, establish a powerful transparency and accountability framework for online platforms, and to foster innovation, growth and competitiveness within the single market.